# Protein Extraction Methods Suitable for Muscle Tissue Proteomic Analysis

**DOI:** 10.3390/proteomes12040027

**Published:** 2024-09-25

**Authors:** Lorenza Vantaggiato, Claudia Landi, Enxhi Shaba, Daniela Rossi, Vincenzo Sorrentino, Luca Bini

**Affiliations:** 1Functional Proteomics Lab., Department Life Sciences, University of Siena, Via Aldo Moro 2, 53100 Siena, Italy; lorenz.vantaggiato2@unisi.it (L.V.); enxhi.shaba@unisi.it (E.S.); luca.bini@unisi.it (L.B.); 2Department of Molecular and Developmental Medicine, University of Siena, Via Aldo Moro 2, 53100 Siena, Italy; daniela.rossi@unisi.it (D.R.); vincenzo.sorrentino@unisi.it (V.S.)

**Keywords:** protein denaturation, muscle tissue, two-dimensional electrophoresis, mass spectrometry, myopathies

## Abstract

Muscle tissue is one of the most dynamic and plastic tissues of the mammalian body and covers different roles, such as force generation and metabolic control. Muscular proteomics provides an important opportunity to reveal the molecular mechanisms behind muscle pathophysiology. To ensure successful proteomic analysis, it is necessary to have an efficient and reproducible protein extraction method. This study aimed to evaluate the efficacy of two different extraction protocols of muscle samples for two-dimensional gel electrophoresis. In particular, mouse muscle proteins were extracted by an SDS-based buffer (Method A) and by a UREA/CHAPS/DTE/TRIS solution (Method B). The efficacies of the methods were assessed by performing an image analysis of the 2DE gels and by statistical and multivariate analyses. The 2DE gels in both preparations showed good resolution and good spot overlapping. Methods A and B produced 2DE gels with different means of total spots, higher for B. Image analysis showed different patterns of protein abundance between the protocols. The results showed that the two methods extract and solubilize proteins with different chemical–physical characteristics and different cellular localizations. These results attest the efficacy and reproducibility of both protein extraction methods, which can be parallelly applied for comprehensive proteomic profiling of muscle tissue.

## 1. Introduction

Muscle tissue is a vital component in the human body, covering more than 50% of body mass. Particularly, skeletal muscle is mainly responsible for generating forces for locomotion and also plays a role in controlling internal organ functions, having endo- and exocrine functions [1]. The muscles release various myokines that arbitrate communication between the muscles and other vital organs and are also involved in muscle proliferation, differentiation and regeneration [2,3]. This highlights the indispensable role of skeletal muscle in organismal health and the pathophysiology of certain chronic diseases, such as diabetes [4], neurodegenerative diseases [5], cardiovascular diseases [6] and cancer [7], as well as various myopathies [8]. The latter encompasses a complex spectrum of disorders characterized by structural and functional abnormalities in the skeletal muscle [9,10,11]. Currently, myopathies present significant clinical challenges, ranging from muscle weakness and cardiac dysfunction to respiratory failure, often leading to profound disability and reduced quality of life of affected individuals [12,13,14]. Understanding the molecular alterations associated with myopathies is crucial for discovering diagnostic biomarkers and therapeutic targets and for developing personalized treatment strategies. Skeletal muscle biopsy remains an important investigative resource for a variety of muscle disorders [15,16]. Moreover, proteomic analyses on muscle tissue have already provided valuable insights into the molecular mechanisms underlying myopathies [17]. Indeed, proteomics offers a comprehensive approach to unraveling the complex protein networks implicated in muscle pathophysiology and to enabling the identification of disease-specific biomarkers and therapeutic targets. Among the analytical techniques employed in proteomics to investigate muscle tissue, two-dimensional electrophoresis (2DE) stands out for its high resolution, reproducibility and capacity to analyze complex protein mixtures, efficiently separating different proteoforms with dynamic post-translational modifications (PTMs). However, the successful application of 2DE relies on effective protein extraction methods that preserve protein integrity, solubility and abundance while minimizing contaminant interference. The optimization of protein extraction protocols is essential for maximizing protein yield and reproducibility and enhancing resolution in 2DE analysis.

Some properties of skeletal muscle tissues make protein extraction and resolution technically challenging. A large number of muscle proteins are highly abundant, such as myosin heavy and light chains, troponins, tropomyosins and actins, and can interfere with the resolution of low abundant proteins showing similar molecular weights (MWs) and isoelectric points (p*I*s). Furthermore, the amounts of integral membrane proteins, which are difficult to solubilize, and proteins with very high MWs, such as nebulin or titin, can be underestimated [18]. The protein extraction methods normally employed for muscle tissue samples include trichloroacetic acid (TCA)/acetone precipitation and some commercially available kits. The efficacy of the latter may be variable depending on the sample type and downstream applications. These methods require tissue homogenization, protein solubilization and contaminant removal. The classical method reported by Hao, R. et al. involves protein precipitation by TCA followed by acetone washing. This procedure has some limitations, including the loss of some specific proteins and the coextraction of contaminants causing horizontal streaking in 2DE gels, making them unsuitable for image analysis [19]. To deal with these challenges, we tested and compared two protein extraction procedures based on two different lysis buffers: the first one was based on detergents such as sodium dodecyl sulfate (SDS) and reducing agents such as dithioerythritol (DTE), with a further process of heating helping protein solubilization and denaturation, preserving protein integrity. The second one was based on UREA as a chaotropic agent, 3-[(3-cholamidopropyl) dimethylammonia]-1-propanesulfonate hydrate (CHAPS) as a detergent, DTE and TRIS as an anti-protease agent. Both methods showed good yields of protein extraction, excellent resolution of the 2D gels that overlapped well and high reproducibility. However, differential analysis of the 2DE gels obtained by the two methods reported a different pattern of protein spot abundance in line with the different extraction procedures. The results stress the importance of using the correct procedure for sample preparation in proteomics.

## 2. Materials and Methods

All procedures including animals were carried out with the utmost care to minimize animal suffering. Isoflurane-anesthetized mice were humanely euthanized by cervical dislocation, following the guidelines approved by the Animal Care Committee of the University of Siena and in compliance with the regulations set forth by the Italian Ministry of Health (64/2020-PR). These procedures adhered to the standards outlined in Directive 2010/63/EU of the European Parliament and the Council of 22 September 2010 concerning the welfare of animals used for scientific purposes. Furthermore, this study was reported in accordance with the ARRIVE guidelines (https://arriveguidelines.org, accessed on 15 January 2024). The experiments were conducted using adult (4 months old) wild-type (WT) animals from strain C57Bl/6J, which were provided with ad libitum access to food and water. The mice were housed under controlled conditions, with a room temperature (RT) maintained between 21 and 25 °C and a relative humidity of 50–60%. The light–dark cycle was set to 12 h. Soleus muscle samples were dissected and immediately frozen at −80 °C until used.

### 2.1. Protein Extraction

Before each type of extraction, 10 mg of soleus samples from three WT mice was divided into two sections using a scalpel. Each section was transferred into a tube and processed by the two different extraction procedures, as described below. Figure 1 shows a simplified flowchart of the two different extraction protocols.

### 2.2. Extraction Method A

Three WT soleus tissues were incubated in 40 µL of denaturation buffer A, containing 2% *w*/*v* SDS and 1% *w*/*v* DTE (pH 6), and then homogenized by using specific beads and TissueLyser II (#85300 Qiagen). The TissueLyser adaptors were frozen before their use at −80 °C to reduce sample heating during the shaking. The samples were agitated for 30 s at 2.5 Hz for 4 times, with a 1 min break after each agitation step. During intervals, the samples were cooled down by placing them in ice and spinning them down at 20,800× *g* for 10 s. After homogenization, the beads were removed and the samples were incubated at 95 °C for 5 min; after that, when they reached RT, they were centrifuged at 20,800× *g* for 15 min, and the supernatants were recovered and underwent acetone precipitation (1:4) overnight at −20 °C. They were then centrifuged at 20,800× *g* for 20 min at 4 °C. The pellet was resuspended in 20 µL of lysis buffer composed of 8 M UREA, 4% *w*/*v* CHAPS and 1% *w*/*v* DTE, and total protein quantification was estimated according to the Bradford method [20].

### 2.3. Extraction Method B

The other parts of the three soleus tissues were directly denatured in 40 µL of denaturation buffer B containing 8 M UREA, 4% *w*/*v* CHAPS, 1% *w*/*v* DTE and 40 mM TRIS (pH 10) and then homogenized by using beads and TissueLyser II (#85300 Qiagen). To maintain similar experimental conditions, the same TissueLyser procedure as described above was performed; i.e., the TissueLyser adaptors were frozen before their use at −80 °C to reduce sample heating, and samples were agitated for 30 s at 2.5 Hz 4 times, with a 1 min break after each agitation step. During the intervals, the samples were cooled down by being placed in ice and spun down at 20,800× *g* for 10 s. After homogenization, the beads were removed, the samples were centrifuged for 15 min at 20,800× *g* at 4 °C, the supernatants were recovered and the total protein concentration was estimated by the Bradford protocol [20].

### 2.4. Two-Dimensional Electrophoresis

All the protein samples extracted by the soleus were separated by two-dimensional electrophoresis (2DE) performed using the Immobiline–polyacrylamide system according to Carleo et al. [21]. In detail, for the analytical runs, a 0.2% (*v*/*v*) carrier ampholyte was added to 60 μg of protein, while for the protein assignment procedures, a 2% (*v*/*v*) carrier ampholyte was mixed with 600 µg of protein; both were diluted in 350 μL of lysis buffer solution and traces of bromophenol blue. The samples were loaded by rehydration on immobilized nonlinear pH 3–10 gradient strips, 18 cm in length (Cytiva, Uppsala, Sweden), and isoelectric focusing (IEF) was carried out utilizing the Ettan™ IPGphor™ Manifold system (Cytiva) at 16 °C with the following voltage program: 0 V for 1 h, 30 V for 7 h, 200 V for 1 h, 300 V for 30 min, a gradient until 3500 V in 2 h, 3500 V for 10 min, from 3500 V to 5000 V in 30 min, 5000 V for 30 min, from 5000 V to 8000 V in 1 h and 8000 V for the rest of the run until reaching a total of 95,000 VhT. After the IEF, focused strips were equilibrated in 6 M UREA, 2% *w*/*v* SDS, 2% *w*/*v* DTE, 30% *v*/*v* glycerol and 0.05 M TRIS-HCL (pH 6.8) for 12 min and a further 5 min with a solution containing 6 M UREA, 2% *w*/*v* SDS, 2.5% *w*/*v* iodoacetamide, 30% *v*/*v* glycerol, 0.05 M TRIS-HCL (pH 6.8) and a trace of bromophenol blue. The second-dimensional separation was performed by posing the strips on the top of the 9–16% SDS gradient polyacrylamide gels (18 cm × 20 cm × 1.5 mm) at 40 mA/a gel constant current at 9 °C until the dye front reached the bottom of the gel. Analytical gels were stained using ammoniacal silver staining [22], while MS-preparative gels were stained following the MS-compatible silver staining protocol [23]. All gels were then digitalized with an Image Scanner III laser densitometer provided with LabScan 6.0 software (Cytiva).

### 2.5. Image and Statistical Analysis

Computer-aided 2D image comparison was carried out using the Melanie 9.0 software (Swiss Institute of Bioinformatics, Quartier Sorge, Batiment Amphipole 1015 Lausanne, Switzerland). To extract quantitative and qualitative differences, all gels from the same sample preparation protocol were matched with their master reference gels, chosen based on resolution and number of spots. Secondly, masters were matched together in inter-class analysis. A statistical ANOVA was also performed by Melanie 9.0 using the percentage of relative volume (%V) (integration of optical density of a single spot (volume) divided by the total volume of spots and expressed as a percentage). Spots were considered differentially abundant when the ratio of the %V means of Methods A and B was more than 3-fold with a statistical *p*-value of < 0.05.

### 2.6. Protein Spot Assignment by Mass Spectrometry

After visualization on MS-compatible silver-stained gels, differentially electrophoretic spots were manually excised and destained in 30 mM potassium ferricyanide and 100 mM sodium thiosulphate anhydrous, then later in 200 mM ammonium bicarbonate before dehydration in 100% acetonitrile (ACN). The spots were rehydrated and digested overnight at 37 °C in trypsin solution (trypsin in 5 Mm ammonium bicarbonate). A total of 0.75 µL of each tryptic digest was placed on the MALDI target, dried, covered with 1 µL of matrix solution composed of 5 mg/mL α-cyano-4-hydroxycinnamic acid (CHCA) dissolved in 50% *v*/*v* ACN and 5% *v*/*v* trifluoroacetic acid (TFA), and dried again. Protein assignment was carried out with an UltrafleXtreme™ MALDI-ToF/ToF instrument (Bruker Corporation, Billerica, MA, USA) armed 200 Hz smartbeam™ I laser, with the following parameters set: 80 ns of delay; ion source 1: 25 kV; ion source 2: 21.75 kV; lens voltage: 9.50 kV; reflector voltage: 26.30 kV; and reflector 2 voltage: 14.00 kV. The applied laser wavelength and frequency were 353 nm and 100 Hz, and the percentage was set to 50%. Spectra were acquired by delayed extraction technology with the reflectron in positive mode and then processed by Flex Analysis software version 3.3 (Bruker). The auto-proteolytic trypsin peptides were used as internal standards to calibrate the acquired spectra. The resulting mass lists were filtered to remove contaminants such as mass matrix-related ions, keratin-derived peaks and trypsin auto-lysis peptide peaks. Protein assignment was carried out by peptide mass fingerprinting (PMF) with the MASCOT search engine, using SwissProt as the database, *mus musculus* as the taxonomy and carbamidomethylation (Cys) and oxidation of methionine as the fixed and variable modifications, respectively, with one missed cleavage allowed and a mass tolerance of 20 ppm. In order to overcome the experimental limitations potentially related to protein assignment, we considered the probabilistic score, the number of experimental matched peptides and the E-value in order to accept assignments. In addition, every missed cleavage site detected by MASCOT in the mass lists was accurately investigated considering the trypsin cleavage limits. Tryptic digests, whose PMF results did not satisfy all the above-mentioned criteria, were excluded. It is to be considered that peaks assigned to sequences were performed by the MASCOT algorithm using a probabilistic method. The mass spectrometry proteomics data have been deposited to the ProteomeXchange Consortium via the PRIDE [24] partner repository with the dataset identifier PXD053117.

### 2.7. Multivariate and Enrichment Analyses

The %V values of total spots per gel were submitted to the XLStat (Addinsoft, 2019) software to perform Pearson’s correlation analysis and unsupervised heatmap analysis to establish the reproducibility of the two extraction methods. The %V values of the statistically significant differential spots were used to execute supervised heatmap analysis, which allowed contemporary differential spot abundance visualization in all gels, clustering the samples (gels) based on Euclidean distance. In order to compare the two different extraction methods, we also highlighted the cellular localizations of the extracted proteins. Therefore, we performed an enrichment analysis of the differential proteins by submitting their UniProt accession numbers (ANs) into the MetaCore software version 6.8 (Clarivate Analytics, Boston, MA, USA), considering the cellular localization obtained by Gene Ontology localization terms.

## 3. Results and Discussion

Skeletal muscle represents the most abundant tissue in mammal bodies, playing a central role in contractile and metabolic functions [25]. Notably, skeletal muscle tissue is peculiarly rich in membrane-associated proteins and high-molecular-weight proteins. Moreover, it is an extremely dynamic tissue that constitutes proteins with extensive PTMs, whose alterations cause the development of some pathologies and dysfunctions [26,27,28,29]. These characteristics determine a difficulty in performing biochemical studies about disease-associated protein alterations and diagnostic or therapeutic biomarker discovery. The 2DE approach, if coupled with an appropriate protein extraction method, is a high-resolution technique essential for resolving and detecting multiple proteoforms [30]. Considering the 2DE properties in revealing muscle tissue molecular details, we tested two different protein extraction methods. To minimize sample variation, soleus samples were taken from three wild-type mice and divided into two sections, each of which was subjected to a different extraction method. Method A comprised a solution containing 2% *w*/*v* SDS and 1% *w*/*v* DTE as a denaturing buffer. In particular, the SDS was used as a strong anionic detergent to disaggregate the highly abundant proteins and to denature proteins rich in hydrophobic residues while the DTE reduced disulfur bridges. The heating process improved protein denaturation and solubilization [31]. Since the SDS was incompatible with the IEF, the samples were subjected to acetone precipitation after protein extraction. Pellets obtained after centrifugation were resuspended in the lysis buffer (8 M UREA, 4% *w*/*v* CHAPS, 1% *w*/*v* DTE) normally used for IEF analysis. CHAPS, a zwitterionic detergent, was substituted for SDS without interfering with the p*I* values of the proteins.

Method B was used to solubilize and denature the proteins of the other section of the soleus muscle samples. It comprised denaturation/solubilization in a UREA-based buffer composed of 8 M UREA, 4% *w*/*v* CHAPS, 1% *w*/*v* DTE and 40 mM TRIS. UREA, a chaotropic agent, breaks hydrogen bonds and other weak bonds, well-solubilizing proteins without producing proteolysis [32]. The DTE was used to reduce disulfide bridges, and the CHAPS, a neutral charged zwitterionic detergent, was used to hydrolyze the hydrophobic and weak bonds of protein–protein interactions, maintaining the individual charge of each protein (p*I*) [33]. TRIS was added to confer a basic pH to the solution, allowing the inhibition of proteases. However, TRIS causes interference in a certain region of 2DE gels, resulting in horizontal streaks; therefore, before sample loading in IEF, the amount of protein sample necessary for the 2DE analysis was diluted in the lysis buffer in a 1:16 ratio.

To obtain highly reliable results, all samples were processed at the same time and all 2DE gels were prepared and used following the same experimental conditions. Protein extraction efficiency was first evaluated based on protein concentration, by a Bradford assay, from about 5 mg of muscle tissue. The protein concentration of each extract is reported in Table 1. Method B extraction recovered a higher number of proteins, with a mean of 21.86 ± 5.27 µg/µL of protein concentration, compared with Method A, with a mean of 8.75 ± 1.72 µg/µL, as shown in Table 1. The differences in protein concentration could be due to the protein loss following the acetone precipitation step required in Method A. Indeed, the loss of protein after acetone precipitation has been largely reported and could be dependent on the protein composition of the sample and the buffer conditions, such as pH, ionic strength and the presence of surfactants [34]. Moreover, Crowell et al. have shown that adding salt (NaCl) in acetone or other ionic species improves the yield of water-soluble proteins.

To evaluate the reproducibility of the extraction methods, we performed a 2DE analysis. Once we acquired the images of the 2DE gels and uploaded them on the Melanie software, we realized that the protein spots on the 2DE gels from Method A numbered 1794, while the protein spots on the 2DE gels from Method B numbered 2513. Also, the numbers of spots on the gels suggested the best yield of protein extraction following Method B. After 2DE gel matching, it was observed that the 2DE gels from the same extraction method were perfectly comparable and the 2DE gels from the two different methods were overlapping, permitting the performance of image analysis. First of all, matching of all gels was performed, extrapolating the %V of all protein spots in all gels. The evaluation of reproducibility of the two methods was tested by Pearson’s correlation analysis, whose coefficients (r) for the spots in Method A and Method B had a mean of r = 0.935 and r = 0.968, respectively (Table 2), confirming that all gels obtained from the same extraction method were significantly correlated (*p* < 0.0001).

Furthermore, a heatmap analysis was performed using the %V of all spots matched in all gels. According to the Euclidean distance, the 2DE gels (samples) were clustered according to method, as shown in Figure 2, corroborating the reproducibility of the 2DE gels.

After image comparison, a differential analysis was performed, highlighting 131 statistically significant (*p* < 0.05) differential abundant spots that increased or decreased more than threefold between Method A and Method B (Figure 3), of which 106 have been assigned to protein species by mass spectrometry.

One hundred and seventeen spots with qualitative differences are shown in the supervised heatmap in Figure 4 and Appendix A, prevalently present in Method B and overall in the low MW range (Figure 5). These findings corroborate the hypothesis that several protein species are lost after acetone precipitation. On the other hand, in Method A, some spots in the high MW range showed enrichment in abundance (Figure 5).

Most of the differential spots assigned to the protein species were referred to muscle tissue, such as troponin, myosin, miozenin 2 and ankyrin, in which it is important to study the proteome of muscular pathologies involving sarcomeres and fibers’ proteins. In particular, many of these proteins are related to some myopathies, such as lethal recessive nemaline and myotonic dystrophies, where mutations in troponin or aberrant splicing specifically affect a particular fiber type [35,36,37]. Furthermore, hypertrophic and dilated cardiomyopathy are primarily caused by monogenic mutations in the myosin 7 gene [38,39]. All the above-mentioned pathologies, some of which are hereditary, are caused by post-translational modifications that can be visualized by two-dimensional electrophoresis. Therefore, the optimized extraction method, enriching this class of proteins, in association with the two-dimensional electrophoresis proteomic approach may be important for the study of these pathologies, allowing a broader and clearer understanding of protein alterations and preventing information loss. This would improve knowledge of the pathogenesis and pathophysiology of cardiomyopathies and skeletal myopathies caused by hereditary and de novo mutations in sarcomeric protein-coding genes, and it would provide new insights into targeted treatment or the development of new early diagnosis techniques. However, these highly abundant proteins mask the low-abundance ones with similar p*I* and MW values, which cannot be considered in the proteomic analysis. Instead, this study showed that Method B improved the number of protein species extractions and, following protein assignment and enrichment analysis, most of these exerted metabolic and kinase functions, such as aldolase, malate dehydrogenase, glyceraldehyde-3-phosphate dehydrogenase, adenylate kinase, etc. Interestingly, there is a growing interest in the role of skeletal muscle in diseases associated with altered metabolism, such as insulin resistance, diabetes and dyslipidemia. In fact, skeletal muscle is the main peripheral site of insulin-stimulated glucose uptake, but it is also considered the primary driver of whole-body insulin resistance [40,41]. Moreover, Method B allowed enrichment in numerous protein species, placed in a low MW range, that, after protein assignment, resulted in protein fragments, such as the dihydrolipoyllysine-residue acetyltransferase component of the pyruvate dehydrogenase complex, the mitochondrial fragment C-term; the L-lactate dehydrogenase A chain fragment C-term; NADH dehydrogenase [ubiquinone] flavoprotein 1, the mitochondrial fragment C-term; malate dehydrogenase, the cytoplasmic fragment C-term; trifunctional enzyme subunit alpha, the mitochondrial fragment C-term and N-term; the troponin T fast skeletal muscle fragment N-term; the Myosin-1 and -7 fragment C-term; ATP synthase subunit alpha, the mitochondrial fragment C-term, etc. Interestingly, Zhang, T. et al. reported in their study that as many protein fragments as ours not only retained their original protein functions but acquired new ones, also by localizing in different subcellular areas [42]. Therefore, these assigned proteoforms are essential for studying specific diseases such as cachexia, dystrophies and sarcopenia, which are associated with altered muscular metabolism [43]. As suggested by the enrichment analysis by GO localization terms in Figure 6, the 2DE patterns obtained by Method B were less enriched in proteins that constitute sarcomeres and muscle fibers than Method A. Furthermore, Method B homogeneously extracted protein species usually localized in the cytosol, mitochondria, extracellular region, vesicles, exosomes, etc. Interestingly, Method B allowed the extraction of several mitochondria membrane proteins, particularly ATPA, ATPB, ECHA, NDUA8 and NDUV1, as suggested by MetaCore enrichment. Notably, there is increasing evidence of associations between mitochondrial alterations of skeletal muscle and lung disease, cancer and other diseases [44,45,46,47]. Also, the loss in extraction of some membrane proteins by Method A could depend on acetone precipitation. Indeed, according to Thongboonkerd et al., acetone leads to a loss of proteins with hydrophobic moieties [48].

In line with our results, the extraction buffer, based on a combination of different solvents and detergents, improved the extraction of proteins with different chemical–physical characteristics, such as solubility and molecular weight. This was also mirrored in the subcellular localization of the proteins. In general, in gel-based or gel-free proteomic analysis, the differences in protein extraction depending on the adopted buffer should be considered. In light of our results, it could be interesting to perform parallel proteomic analyses of the muscular tissues, using the two different methods, to better understand the alteration of protein patterns in muscular pathologies.

## 4. Conclusions

The optimization of protein extraction protocol is essential to ensure successful and reproducible proteomic analysis. This work presents two highly efficient and reproducible protocols for muscle tissue proteomic analysis. Method A was based on an extraction buffer containing a strong anionic detergent such as SDS, which needs protein precipitation before 2DE analysis, while by Method B, proteins were extracted thanks to a buffer consisting of 8 M UREA, 4% *w*/*v* CHAPS, 1% *w*/*v* DTE and 40 mM TRIS. In comparison with the other, extraction Method B was simpler and faster, reporting the least protein loss. However, following 2DE gel analysis, the two protein extraction procedures could be considered complementary, highlighting different patterns of protein abundance. In particular, Method A showed a better protein extraction yield for heavy-molecular-weight protein species, which were mostly contractile fibers and sarcomere proteins. Conversely, Method B homogeneously extracted cytosolic, extracellular, mitochondrial and exosomal proteins and several protein fragments. The observed differences in protein extraction, depending on the extraction buffer adopted, should be considered when gel-based or gel-free proteomic analysis is performed. The findings from this study can be applied to the comprehensive proteomic profiling of muscle tissue, facilitating the identification of disease-associated protein alterations and biomarker discovery in muscular pathologies.

## Figures and Tables

**Figure 1 proteomes-12-00027-f001:**
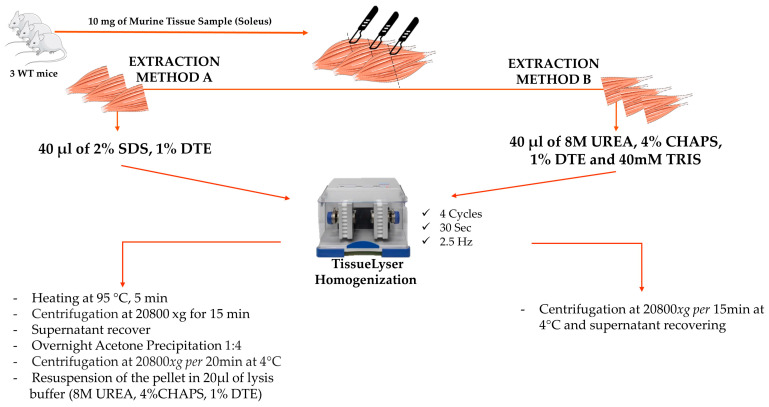
Flowchart of the two different extraction protocols. A total of 10 mg of soleus samples from three wild-type mice was divided into two sections using a scalpel. Each section obtained from the same soleus sample was processed by a different extraction method. Method A comprised the extraction buffer containing 2% *w*/*v* SDS and 1% *w*/*v* DTE and homogenization by TissueLyser II (Qiagen, Hilden, Germany). After homogenization, the samples were incubated at 95 °C for 5 min, and after centrifugation at 20,800× *g* per 15 min at RT, the obtained supernatant underwent overnight cold acetone precipitation (1:4) at −20 °C. After centrifugation at 20,800× *g* per 20 min at 4 °C, the pellets were recovered and resuspended in 20 µL of lysis buffer composed of 8 M UREA, 4% *w*/*v* CHAPS and 1% *w*/*v* DTE. Method B comprised the extraction buffer containing 8 M UREA, 4% *w*/*v* CHAPS, 1% *w*/*v* DTE and 40 mM TRIS and homogenization by TissueLyser II; the solubilized proteins were recovered in the supernatant after centrifugation at 20,800× *g* per 15 min.

**Figure 2 proteomes-12-00027-f002:**
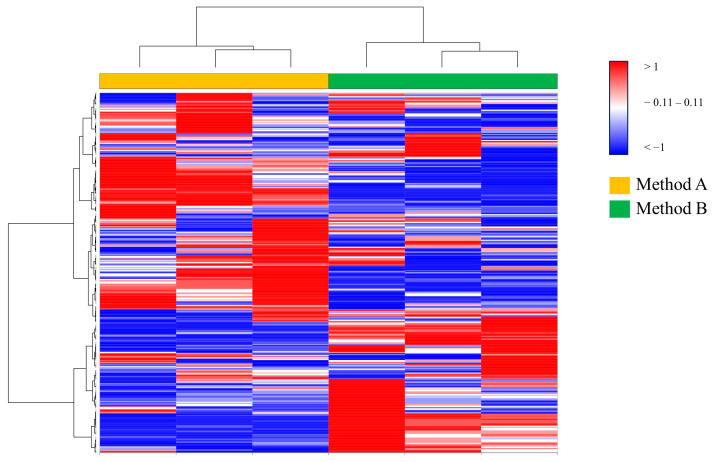
Unsupervised heatmap analysis was performed on all matched spots in all gels. A color change from blue to red indicates lower or higher protein abundance, respectively. Each row corresponds to a protein spot, while each column corresponds to an individual gel sample. The dendrogram on the top shows the gels clustered according to the corresponding extractions: Method A (yellow) and Method B (green).

**Figure 3 proteomes-12-00027-f003:**
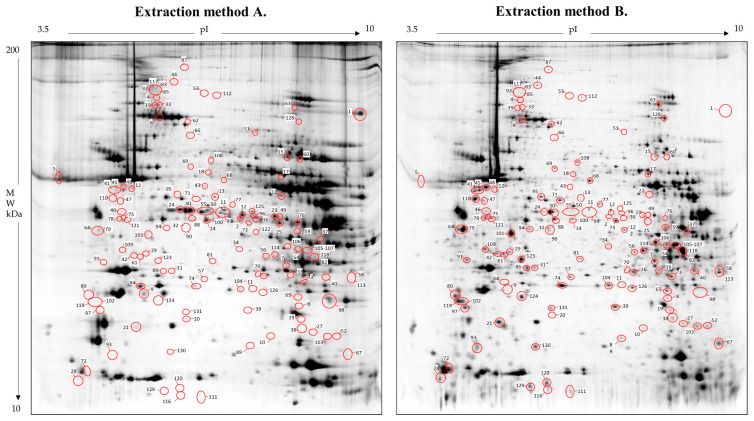
Reference gel maps for extraction Methods A and B. The red circles represent the statistically relevant spots, with a fold change ratio of ≥ 3 between the two extraction methods. Each number is associated with a specific protein species, reported in Appendix A.

**Figure 4 proteomes-12-00027-f004:**
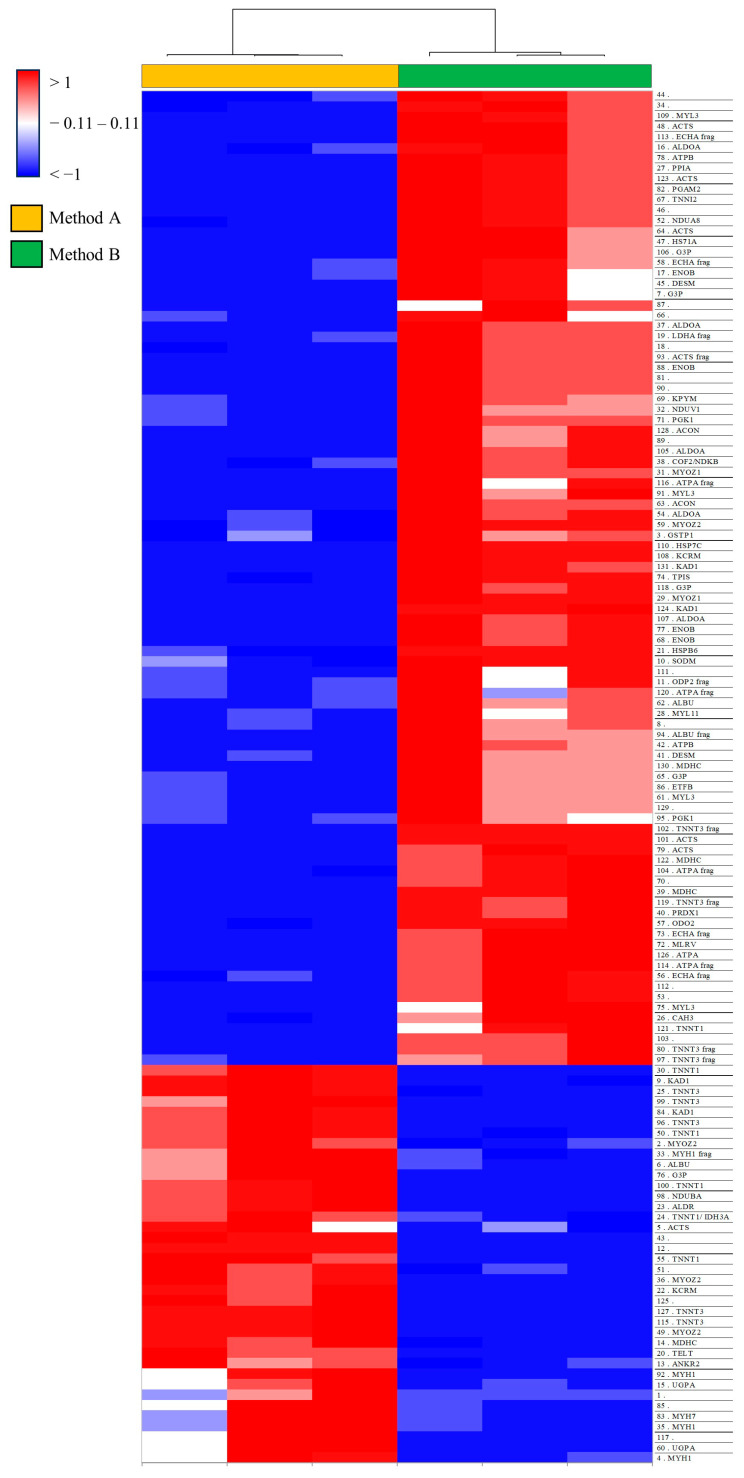
A supervised heatmap analysis was performed on significant differential spots between Methods A and B. A color change from blue to red indicates lower or higher protein abundance, respectively. Each row corresponds to a differential protein spot, while each column corresponds to an individual 2DE gel (sample). If assigned to a protein species, the row corresponding to the differential protein spots reports the UniProt entry name.

**Figure 5 proteomes-12-00027-f005:**
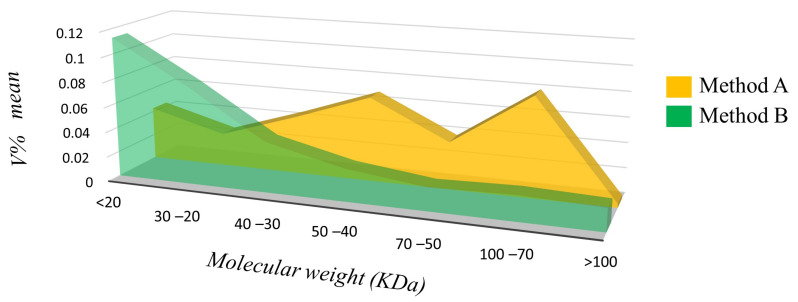
This graph shows that the amount of differentially abundant spots localized along the ranges of molecular weight relies on the method of extraction. Method A enriched higher-molecular-weight protein spots while Method B enriched protein spots at lower molecular weights.

**Figure 6 proteomes-12-00027-f006:**
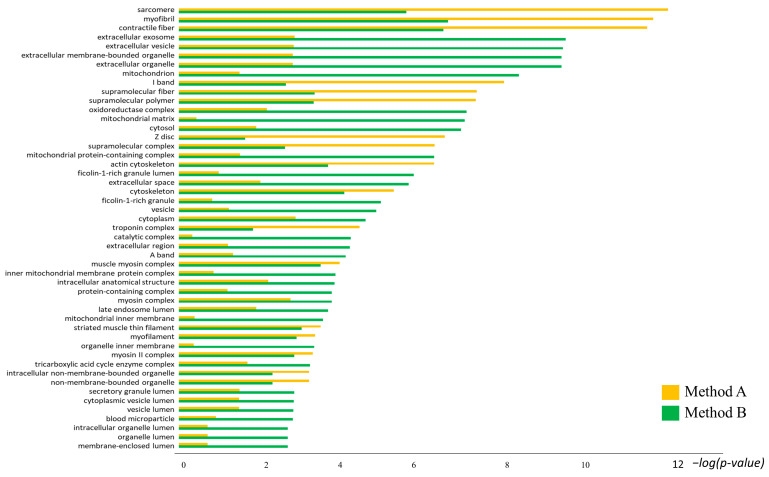
Histograms reporting the comparisons of GO localization terms of up-regulated proteins in extraction Methods A and B.

**Table 1 proteomes-12-00027-t001:** Protein concentration obtained for each extraction by Bradford assay.

	Method A (µg/µL)	Method B (µg/µL)
Sample 1	10.5	26.8
Sample 2	8.7	16.3
Sample 3	7.05	22.5

**Table 2 proteomes-12-00027-t002:** Pearson’s correlation matrix was performed on the base %V of all matched spots for each extraction method. The numbers reported in the tables correspond to the Pearson correlation coefficients’ (r) mean of all spots correlated with themselves on the other gels obtained from the same method of protein solubilization/denaturation. All gels obtained from the same extraction method are significantly correlated (*p* < 0.0001).

Pearson’s Correlation Matrix (r)							
Extraction Method A				Extraction Method B			
Gels	Sample 1	Sample 2	Sample 3	Gels	Sample 1	Sample 2	Sample 3
Sample 1	1	0.935	0.93	Sample 1	1	0.97	0.965
Sample 2	0.935	1	0.941	Sample 2	0.97	1	0.971
Sample 3	0.93	0.941	1	Sample 3	0.965	0.971	1

## Data Availability

The mass spectrometry proteomics data have been deposited to the ProteomeXchange Consortium via the PRIDE partner repository with the dataset identifier PXD053117.

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
