# Peer review of "Protein Extraction Methods Suitable for Muscle Tissue Proteomic Analysis"

_proteomes, 2024, doi:10.3390/proteomes12040027_

Round 1
Reviewer 1 Report
Comments and Suggestions for Authors
The study titled “Protein Extraction methods suitable for muscle tissue proteomic analysis” presents a comparison of two protein extraction methods for proteomic analysis of skeletal muscle tissue, offering valuable insights into their respective strengths and limitations. The manuscript is well-written and detailed, covering all necessary aspects of a reproducible protocol for muscle tissue proteomic analysis. The thorough experimental design, combined with detailed quantitative and qualitative analysis, provides a solid foundation for understanding the proteomic differences elicited by each method. This level of detail is crucial for reproducibility in proteomic studies. However, the utility of these methods in the identification of novel or low abundance proteins are not discussed in the manuscript. Hence this work seems to be like a meagre comparison between the methods. The authors also do not report the application of these methods in other types of muscles narrowing down the benefit of such extraction methods. Addressing the suggested major and minor revisions would significantly enhance the clarity, impact, and applicability of the study, making it a valuable contribution to the field of proteomics.
Major revisions
1. The focus on soleus muscle samples from wild-type mice limits the generalizability of the results. Including additional muscle types or disease models would enhance the applicability of the findings to broader contexts
1. While Table 3 lists the identified proteins, the manuscript lacks a detailed explanation of the proteins associated with each extraction method. Moving Table 3 to a supplementary document and providing a comprehensive discussion on the protein classes and their implications would enhance the clarity and utility of the findings.
3. Though the authors mention that the extraction method B yielded more proteins than method A, the 2-DE gel image (Figure 3) shows an increased amount of protein abundance and spots.
4. While the study acknowledges the limitations of acetone precipitation in Method A, which leads to protein loss, it does not explore potential modifications or alternatives that could mitigate this issue. This would be expected as the current manuscript is a method-oriented one
5. The discussion could delve deeper into the broader implications of the findings, particularly in terms of how these extraction methods could be applied to clinical or diagnostic settings. More emphasis on the potential for biomarker discovery and therapeutic development would strengthen the relevance of the study.
Minor revisions:
1. Correct minor typographical errors (e.g., "“solubilizzation instead of solubilization” in line 75 page 2), “ageyient instead of agent”, "performed" instead of "it was performed" in line 268 page 7).
2. Typographic errors in the legend of Figure 2, the heatmap depicted in Figure 4 has incomplete accession numbers. Ensure that the legends for figures are complete and accurate. Inconsistencies in representation should be addressed for clarity.
3. The column labeled "FC" should be clarified as "Fold-Change" with appropriate formatting (use of decimal points instead of commas).
Comments on the Quality of English LanguageMinor typographical and grammatical errors are to be addressed.
1. Correct minor typographical errors (e.g., "“solubilizzation instead of solubilization” in line 75 page 2), “ageyient instead of agent”, "performed" instead of "it was performed" in line 268 page 7).
2. Typographic errors in the legend of Figure 2, the heatmap depicted in Figure 4 has incomplete accession numbers. Ensure that the legends for figures are complete and accurate. Inconsistencies in representation should be addressed for clarity.
Author Response
Comments in response to the Reviewer 1 are reported in the file below.

Reviewer 2 Report
Comments and Suggestions for Authors
Comments and suggestions are attached in pdf file

Author Response
Comments in response to the Reviewer 2 are reported in the file below.

Reviewer 3 Report
Comments and Suggestions for Authors
This manuscript compared two frequently used protein extraction methods and applied them two in muscle tissue protein extraction. They found that both methods are good.
Unfortunately, I don’t see any innovation in this manuscript. If they applied the two methods to extract proteins in other tissues, they may find both methods are also okay.
Author Response
Comments in response to the Reviewer 3 are reported in the file below.

Reviewer 4 Report
Comments and Suggestions for Authors
This work aims to evaluate two different methods for protein extraction from muscle tissue, one utilising acetone extraction, and the other just using the same buffer that will be required for analysis later. Three soleus muscles from mice were divided in two, with the two halves undergoing protein extraction by the two different protocols. The resulting samples were analysed by 2D gel electrophoresis and mass spectroscopy. Results revealed a higher protein yield, and a larger number of unique protein spots, in the simpler protocol. It was also more effective at extracting low molecular wright proteins, while the acetone extraction method was more effective in extracting high molecular weight proteins. Identification of the proteins by mass spectroscopy revealed that the acetone extraction is more effective at extracting sarcomeric proteins, while the simpler method yields proteins from a wider variety of subcellular compartments.
It’s unclear why these were the two extraction methods considered. But once the two methods were selected, the manuscript is effective at comparing how well they extract proteins for proteomics analysis.
- Clarify how method A was chosen/developed, as it doesn’t seem to be exactly the same as any of the methods cited.
- The manuscript is inconsistent if 20 or 40 uL of buffer were used to incubate and lyse each sample in.Please clarify and ensure consistency.
- In lines 115-116, the exact composition and pH of the denaturing buffer are missing.
- In line 128, the pH of the buffer is missing.
- Please provide a reference for the Bradford method.
- The manuscript is inconsistent on how precise protein concentration determination is using the Bradford method – line 126 says protein concentration was determined, while in line 137 it was only estimated. Please give the uncertainty – for instance, as standard deviation – in the Results section (Table 1), as that will allow the reader to find out if the concentration was determined or just estimated.
- In line 177, the composition of trypsin solution needs to be given.
- In lines 252 and 253, a mean number of spots is given. Please give also the standard deviation or range.
- Why is the x axis scale go from higher to lower values (from left to right) in figure 5?
Comments on the Quality of English Language
- English needs checking and rewriting – there are plenty of small mistakes in both spelling and grammar, including the use of a comma instead of a dot as the decimal point. At some points, the selection and order of words makes more sense in Italian than in English.
- Formatting is also inconsistent, for instance the space between number and unit is missing in a number of instances, and mM cannot be w/v.
Author Response
Comments in response to the Reviewer 4 are reported in the file below.

Round 2
Reviewer 3 Report
Comments and Suggestions for Authors
Unfortunately, I don’t see any innovation in this manuscript. One can download other protein extraction methods as well.
Author Response
We understand your point of view and we take note of it. At this point we will wait for Editor decision since there is nothing of constructive to reply but it is only your opinion. Thank you